# Kidney Health and Care: Current Status, Challenges, and Developments

**DOI:** 10.3390/jpm13050702

**Published:** 2023-04-22

**Authors:** Ming-Yen Lin, Yi-Wen Chiu, Yu-Hsuan Lin, Yihuang Kang, Ping-Hsun Wu, Jeng-Huei Chen, Hsing Luh, Shang-Jyh Hwang

**Affiliations:** 1Division of Nephrology, Department of Internal Medicine, Kaohsiung Medical University Hospital, Kaohsiung Medical University, Kaohsiung 807, Taiwan; 1080421@kmuh.org.tw (M.-Y.L.); chiuyiwen@kmu.edu.tw (Y.-W.C.); 970392kmuh@gmail.com (P.-H.W.); 2Taiwan Instrument Research Institute, National Applied Research Laboratories, Hsinchu 300, Taiwan; 3Department of Information and Management, National Sun Yat-Sen University, Kaohsiung 804, Taiwan; ykang@mis.nsysu.edu.tw; 4Department of Healthcare Administration and Medical Informatics, Kaohsiung Medical University, Kaohsiung 807, Taiwan; 5Department of Mathematical Sciences, National Chengchi University, Taipei 116, Taiwan; jhchen@nccu.edu.tw (J.-H.C.); slu@nccu.edu.tw (H.L.); 6Graduate Institute of Medicine, College of Medicine, Kaohsiung Medical University, Kaohsiung 807, Taiwan; 7Institute of Population Health Sciences, National Health Research Institutes, Zhunan Town, Miaoli County 350, Taiwan

**Keywords:** kidney care, healthcare system, digital health, health equity, sustainability

## Abstract

The concept of chronic kidney disease (CKD) originated in the 2000s, and an estimated 850 million patients are currently suffering from health threats from different degrees of CKD. However, it is unclear whether the existing CKD care systems are optimal for improving patient prognosis and outcomes, so this review summarizes the burden, existing care models, effectiveness, challenges, and developments of CKD care. Even under the general care principles, there are still significant gaps in our understanding of the causes of CKD, prevention or care resources, and care burdens between countries worldwide. Receiving care from multidisciplinary teams rather than only a nephrologist shows potential profits in comprehensive and preferable outcomes. In addition, we propose a novel CKD care structure that combines modern technologies, biosensors, longitudinal data visualization, machine learning algorithms, and mobile care. The novel care structure could simultaneously change the care process, significantly reduce human contact, and make the vulnerable population less likely to be exposed to infectious diseases such as COVID-19. The information offered should be beneficial, allowing us to rethink future CKD care models and applications to reach the goals of health equality and sustainability.

## 1. Introduction

Chronic kidney disease (CKD) was conceptualized by the National Kidney Foundation Kidney Disease Outcomes Quality Initiative guidelines in 2002 and is mainly determined by kidney function impairments caused by different injuries [1]. For convenient communication, the glomerular filtration rate and urine protein have been widely used to determine CKD in clinical care and research. Kidney function is usually estimated based on the glomerular filtration rate through a mathematical equation inputting demographic factors and serum creatinine results, whereas the urine albumin creatinine ratio mainly determines urine protein. Based on the results of albuminuria (three categories) and estimated glomerular filtration rate (eGFR) (six classes) constructing risk matrix, subjects are classified into 1-5 CKD stages and evaluated as being at low, moderately increased, high, or very high risk of poor prognosis. It is estimated that about 850 million people worldwide have CKD or are in the low- to high-risk category, progressing to worsening kidney function [2]. Patients in more advanced CKD stages or high-risk states are more likely to suffer from cardiovascular disease, end-stage kidney disease, and mortality.

The care and management of CKD patients is more challenging than for other common chronic diseases, mainly when CKD patients have pre-existing chronic conditions or new developing complications. A Canadian population-based study showed that over 70% of CKD patients have at least one comorbidity, with hypertension and diabetes mellitus, which account for 46.6% and 17.8%, being the most common [3]. In addition, the common complications caused by kidney failure (e.g., anemia, metabolic acidosis, electrolyte disorders) usually bring CKD care in line with more complicated situations. The 2012 Kidney Disease Improving Global Outcomes clinical practice guidelines strongly recommend that patients should receive nephrology care when their eGFR deteriorates to below 30 mL/min/1.73 m^2^, or no later than one year before dialysis [4]. However, the optimal CKD care compositions within one healthcare system remained indeterministic. The CKD care models have been the subject of previous systematic reviews and meta-analyses, showing that the specialist compositions within the CKD care models vary between studies and may be affected by different disease management concepts and resource distributions [5]. Therefore, it is necessary to understand and choose an optimal CKD care model based on the disease burden and healthcare resources. This narrative review summarizes the CKD care burdens, existing CKD care models and missions, their effectiveness and challenges, and developments.

## 2. CKD Prevalence and Etiology

The global CKD prevalence is around 10–12% worldwide [2], and CKD etiology varies across geographical regions. The main causes of CKD differences between countries may affect prevention and healthcare resource allocations. Chronic conditions relevant to kidney diseases, such as hypertension and type 2 diabetes mellitus (T2DM), rapidly accelerated in most developing and developed countries. The United States has reported a relatively stable T2DM prevalence (nearly 40%) in adults with CKD across the last decade [6]; however, some developing countries also face uncommonly caused nephropathy that will impact CKD care loads. In India’s southern and northern areas, unknown etiology accounted for 50% of CKD prevalence from 2000 to 2014 [7]. Heat stress has been considered the most likely cause of kidney disease in these regions. Still, other factors such as heavy metals, agrochemicals, infectious diseases, and tobacco and betel leaf use are also suspect [8]. Even in countries with good sanitary conditions, infectious disease threats to kidney health are still essential to pay attention to. In 2018, patients with hepatitis C infection treated with antivirus agents accounted for 7% of the Taiwanese hemodialysis population [9]. Future studies should determine how to accurately estimate the causal effect of CKD etiology and rank their influences for a precise preventive strategy within one country.

## 3. CKD Care Composition and Missions

Regardless of the etiology of kidney disease, kidney care highly depends on the healthcare worker’s knowledge, abilities, and skills. Traditional CKD care involves standard composite workforces, including nephrologists, nurses, pharmacists, and dieticians [5]. A recent review suggests that the nephrologist service is the ideal component within one CKD care system; however, health workers or primary care providers are necessary community human resources, particularly in low- and middle-income countries [10]. It is not surprising that common causes of CKD such as hypertension, diabetes, and obesity affect a large proportion of the population, requiring more healthcare professionals experienced in underlying disease management and early detection of CKD. Taiwan developed the CKD care system in 2003. We used this example to represent the care composition and the main CKD care objectives (Figure 1). The care system incurred various missions to cover CKD patients who may be at different risks of complications, kidney failure, and death. Patients with risk factors for CKD should routinely receive CKD screening to detect CKD at early stage, which could improve public awareness of CKD. In early phase CKD, health workers should assist patients in self-care management: monitoring blood pressure, lifestyle modification, managing their medications, and diet control, which is essential for CKD management. Appropriately regular clinical kidney function assessment—eGFR and proteinuria follow-up—is crucial for early detection of CKD progression and possible precipitating factors for CKD progression. When patients’ kidney function falls below 30 mL/min/1.73 m^2^ or proteinuria is uncontrolled, nephrology care is strongly advised to obtain a proper evaluation for the reversible etiology of declining eGFR and to adjust CKD management as needed [4]. In addition, the optimal timing for shared decision making for patients with stage 5 is required to ensure patient preferences and clinical benefits when choosing renal replacement therapy modality and maturing dialysis access. Noteworthy, Taiwan’s universal pay-for-performance CKD care program has incorporated pharmacist services and covered patients with acute kidney diseases and disorders in the early 2020s. The health insurance also reimburses Chinese herbal medicine physicians providing care services to patients with CKD stages 2–5. In summary, CKD care adheres to the ultimate goal of human health, which is to promote well-being and prolong life, and needs support from different health specialists based on life levels and affordability.

## 4. Effectiveness Assessment in CKD Care Model

Various structured CKD care models in the early phase (1992–2014) showed some benefits in kidney, cardiovascular, and mortality outcomes, but the benefits of multidisciplinary care (MDC) are not consistent in all randomized controlled trials [5]. On the other hand, one more recent review supports that MDC helps slow the progression of CKD [11]. This discrepancy may be attributed to study design, patient characteristics, care continuity, sample size, and study duration. Due to limited resources, results from those randomized clinical trial studies may not reflect the overall reality. It is noteworthy that results from one longer randomized clinical trial study have shed new light on the effectiveness of the care model in improving kidney outcomes (range of point estimation of hazard ratio: 0.45–0.81) and mortality (point estimation of hazard ratio: 0.85) [12]. In addition, our previous study successfully demonstrated a lower long-term, age-standardized incidence rate of dialysis after implementing a universal MDC model [13]. Recent findings suggest governments should draw up appropriate CKD care policies and arrange relevant priorities based on their self-socioeconomic conditions.

## 5. Challenges of CKD Care

Although different care models have shown promise in improving patient outcomes, the current CKD care system faces several challenges. Appropriate screening/early detection of CKD is somewhat controversial. Regarding the early detection of patients with progressive disease, we probably need good, friendly used prediction tools to help identify patients at risk of progression to ESKD. Only 65% of countries recognize that serum creatinine without eGFR reporting could usually or always be available in primary care [14]. In other words, nearly half of the world’s countries face insufficient or even a lack of critical tools for primary CKD care, even though they already have the workforce. Second, there is a lack of consistent, practical methods for identifying the primary causes of CKD. Although diabetes is considered the leading cause of CKD in most high-income countries, classical diabetes nephropathy is less proven by biopsy. Still, most other CKD etiologies in high-income countries depend on clinical signs rather than biopsy findings, and a severe lack of resources restricts many low-income countries. In a Japan CKD cohort report, 38% of patients were diagnosed with diabetes, but only 6% who underwent a kidney biopsy were diagnosed with classic diabetes nephropathy [15]. Without kidney biopsy information, the cause of CKD may be incorrect, causing care resources to be misallocated and hampering CKD prevention efforts. The next challenge is the rapidly increasing care burden in CKD. Unsurprisingly, CKD patients live with more illnesses when care improves life expectancy. A recent Canadian study has revealed that a significant proportion of patients with CKD were living with multiple comorbidities. Specifically, 25% of patients with CKD had three or more comorbid conditions, and an even more concerning 7% had more than five chronic conditions [3]. It is reasonable to manage and control similar metabolic diseases co-existing with CKD. However, other comorbidities, such as infectious diseases, malignancies, mental illness, or dementia, may exacerbate healthcare workers’ burdens. The healthcare system must consider integrating care knowledge, authorities, and responsibility between different care specialists. There are still significant gaps in how to reach care targets between healthcare professionals, one patient, and his/her family. The unequal kidney care workforce distribution between countries is another critical issue. Despite the fact that nephrologists play a significant role in kidney failure care in over 89 percent of countries, the number of nephrologists in low-income countries remains insufficient [16]. In many low-income countries, there is less than one nephrologist for every million people with CKD; however, in more than half of high-income countries, less than 40,000 people with CKD could share one nephrologist [15]. The lack of cultivation and certification systems is a reason for the shortage of nephrologists in low-income countries, but they can be trained abroad by establishing regional training hubs. Those who have undergone advanced training in high-income countries are expected to be pioneers in improving nephrology care in their countries of origin. For example, Tanzania has successfully improved its nephrology care capacity through the above approach to reflect the gradually increasing CKD burden [17]. The international societies of nephrology urgently need to establish a novel platform for international cooperation, which governments and businesses can use to advance kidney health and care everywhere in the world.

## 6. Perspectives on Future Kidney Care

From 2020–2022, the COVID-19 pandemic impacted kidney care, particularly for vulnerable CKD patients. A dilemma in nephrology is how to strike a balance between reducing the risk of infection and providing uninterrupted care. Fortunately, several decision-supportive modules have been posed that could assist CKD care and may reduce the risk of patients contacting infected cases in hospitals. The U.S. Bipartisan Budget Act allows Medicare to reimburse telecare twice seasonally for patients undergoing dialysis in 2018 [18]. In addition, with the popularity of smartphones, many nutritional apps have been used to assist CKD patients with health management [19]. Therefore, CKD care that integrates and interacts with patients and instruments (biosensors), information (visualization for longitudinal changes), algorithms (artificial intelligence), and physicians (decision) should be developed. Such a care system should comprise four elements: a biosensor system, a longitudinal data visualization system, a machine-learning system, and an expert decision system (Figure 2). The repeated care flow consequently produces data to information (sensor to meaningful information), information to intelligence (understanding how and why information makes sense to human thinking), and intelligence to a decision (judging model prediction by expert knowledge). The biosensors could continuously or periodically measure key physiological parameters such as heart rate, blood pressure, blood sugar, and respiration, thereby improving unnecessary medical visits. For example, over 15% of CKD patients may experience heart failure, which could incur excessive hospitalization and mortality risks [20]. It is possible to develop a rule advising patients at home to call for emergency medical help when biosensors determine they are experiencing life-threatening heart failure. The judged criteria should be formed by integrating information from different biosensors, including (but not limited to) shortness of breath, rapid or irregular heartbeat, and high B-type natriuretic peptide and potassium. A recent review article synthesized the effects of telecardiology in low-income and developed countries. It showed the tremendous potential benefits of telemedicine in early diagnosis, early treatment of cardiovascular disease, mortality reduction, and economic saving [21]. The economic effect of telemedicine on cardiovascular disease includes a traveling reduction, days off work reduction, number of visits reduction, and a decrease in patient referrals for assessment in specialized medical centers. Based on the above evidence, personalized biosensor technology should bring CKD care into a new ambit.

Trajectory patient data changes are more informative than cross-sectional measures and should be widely adopted for future CKD care. Machine learning approaches that generate accurate and precise predictions from complex data structures have shown promise in erythropoiesis-stimulating agent prescription in hemodialysis [22]. It is expected that more human-like predictions and applications in kidney care will be developed and evaluated.

Care challenges are a blessing in disguise when seeking ideal solutions. Although the four subsystems are distinct, they are continuously cross-mapping by communicating and learning from one another. This novel structure, combined with wireless communication technology and a drug logistics platform, provides an appropriate, low human contact, trustworthy, burden-sharing, and sustainable kidney care environment.

## 7. Conclusions

This review summarized contemporary challenges and future developments in kidney care. These perspectives do not cover all elements of kidney care in great detail but could promote future healthcare system improvement to achieve equality and sustainability.

## Figures and Tables

**Figure 1 jpm-13-00702-f001:**
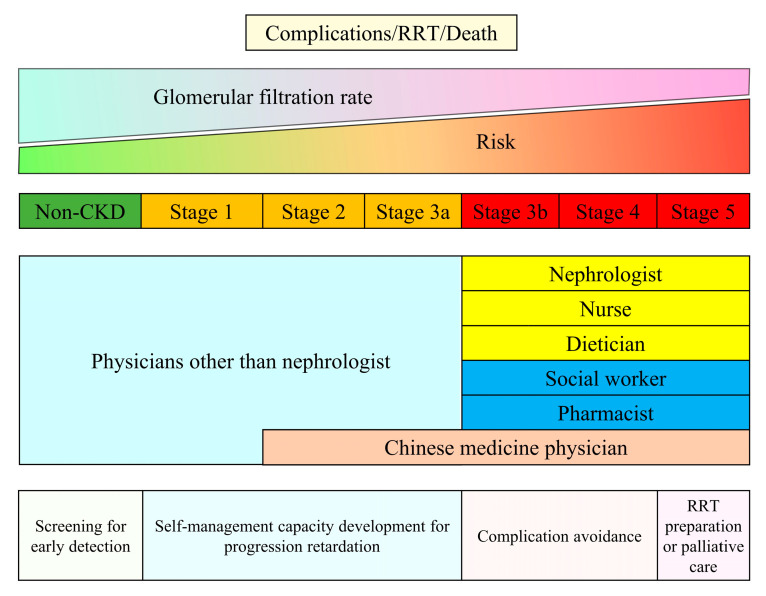
Mission and compositions of the Taiwan Chronic Kidney Disease Care Model. Footnote: The Pre-ESRD and Early CKD care programs are universal pay-for-performance reimbursement systems launched in 2006 and 2011. Physicians other than nephrologists who received courses from the Taiwan Society of Nephrology could provide reimbursed care for patients with CKD earlier than stage 3a. In the early 2020s, the Pre-ESRD care program also incorporated pharmacist services and was extended to patients with acute kidney diseases and disorders. In addition, the Taiwan National Health Insurance passed a new act to reimburse Chinese herbal medicine physicians providing intensive care services to patients with late CKD in 2021. Abbreviation: CKD—chronic kidney disease; RRT—renal replacement therapy.

**Figure 2 jpm-13-00702-f002:**
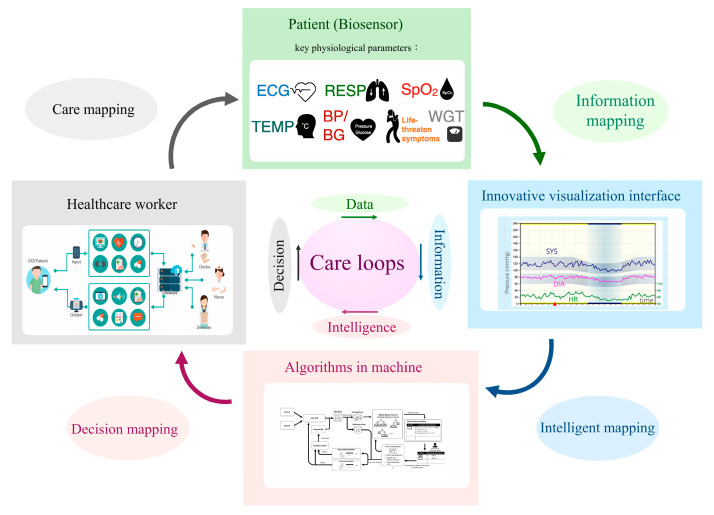
A novel structure for future kidney care. Footnote: Abbreviation: ECG—electrocardiography; RESP—respiration; SpO_2_—saturation of peripheral oxygen; TEMP—temperature; BP—blood pressure; BG—blood glucose; WGT—weight; SYS—systolic blood pressure; DIA—diastolic blood pressure; HR—heart rate. All pictures within the rectangle represent examples of each subsystem.

## Data Availability

Not applicable.

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
