# Peer review of "Kidney Health and Care: Current Status, Challenges, and Developments"

_jpm, 2023, doi:10.3390/jpm13050702_

Round 1
Reviewer 1 Report
Major:
1. The manuscript would greatly benefit from editing.
2. The four subsystems are great tools if applicable to monitor CKD patients. However, it was very simplistically designed and insufficiently explained by the authors. For example, it is not clear what type of biosensor would measure the wide range of physiological parameters suggested by the authors. How will this be implemented, especially in low-income countries, including patients’ compliance, education, training, and potential error of parameter measurements. How will this system decrease the finical burden from the government sector, private sector, and patient point of view.
Minor:
Line 72: Please indicate if the prevalence is worldwide or regional specific.
Line 134: Please use the abbreviation properly throughout the manuscript (e.g., multidisciplinary (MDC)).
Line 164 and 182: The authors may have written this sentence better; please rewrite, “According to Canada's systematic analyses, 25% and 7% of adults with CKD had three and more and five and more comorbidities, respectively”, and “International nephrology societies must create a novel platform for international cooperation involving governments and businesses to promote kidney health and care for world everyone.”
The manuscript would greatly benefit from editing
Author Response
Major:
Question 1. The manuscript would greatly benefit from editing.
Answer to question 1: Thank you for the helpful editing. Moreover, Proofreading Service UK did the English edits to ensure its quality.
Question 2. The four subsystems are great tools if applicable to monitor CKD patients. However, it was very simplistically designed and insufficiently explained by the authors. For example, it is not clear what type of biosensor would measure the wide range of physiological parameters suggested by the authors. How will this be implemented, especially in low-income countries, including patients' compliance, education, training, and potential error of parameter measurements. How will this system decrease the finical burden from the government sector, private sector, and patient point of view.
Answer to question 2: Thank you for the valuable suggestions. We added some descriptions highlighting the biosensor's possible applications in CKD care. Based on limited scientific evidence on the prosed framework, the description below poses the possibilities of applications rather than offering comprehensive viewpoints from different sectors.
For example, over 15% of CKD patients may occur heart failure, which could incur excessive hospitalization and mortality risks [19]. It is possible to develop a rule advising patients at home to call emergency medical help when biosensors determine they are experiencing life-threatens heart failure. The judged criteria should be formed by integrating information from different biosensors, including but not limited to shortness of breath, rapid or irregular heartbeat, and high B-type natriuretic peptide and potassium. A recent review article synthesized the effects of telecardiology in low-income and developed countries. It showed the tremendous potential benefits of telemedicine in early diagnosis, early treatment of cardiovascular disease, mortality reduction, and economic saving [20]. The economic effect of telemedicine on cardiovascular disease includes a traveling reduction, days off work reduction, number of visits reduction, and a decrease in patient referrals for assessment in specialized medical centers. Based on the above evidence, personalized biosensor technology should bring CKD care into a new ambit.
Minor:
Question 3. Line 72: Please indicate if the prevalence is worldwide or regional specific.
Answer to question 3: Thank you for the suggestion. We have revised the sentence: Global CKD prevalence is around 10–12% worldwide [2], and CKD etiology varies across geographical regions.
Question 4. Line 134: Please use the abbreviation properly throughout the manuscript (e.g., multidisciplinary (MDC)).
Answer to question 4: We have added the full name, multidisciplinary care, and abbreviation (MDC) when they first appeared (Line 132) in the new edition. We have also applied the abbreviation throughout the manuscript.
Question 5. Line 164 and 182: The authors may have written this sentence better; please rewrite, "According to Canada's systematic analyses, 25% and 7% of adults with CKD had three and more and five and more comorbidities, respectively", and "International nephrology societies must create a novel platform for international cooperation involving governments and businesses to promote kidney health and care for world everyone."
Answer to question 5: We have edited the first sentence: "A recent Canadian study has revealed that a significant proportion of patients with CKD were living with multiple comorbidities. Specifically, 25% of patients with CKD had three or more comorbid conditions, and an even more concerning 7% had more than five chronic conditions.". The second one has been rewritten as "It is urgent that the international societies of nephrology establish a novel platform for international cooperation, which governments and businesses can use to advance kidney health and care everywhere in the world.".
Reviewer 2 Report
This is an interesting narrative review about the involvement of healthcare systems in the care of patients with CKD. I only have some minor comments:
- Line 82: “Notably, kidney disorders caused by infectious infections…”. I suggest correcting this, it sounds redundant.
-Line 134: MDC is used, I suggest explaining this acronym when firstly used (multidisciplinary care?)
Author Response
Comments and Suggestions for Authors
This is an interesting narrative review about the involvement of healthcare systems in the care of patients with CKD. I only have some minor comments:
Question 1- Line 82: “Notably, kidney disorders caused by infectious infections…”. I suggest correcting this, it sounds redundant.
Answer to question 1: Thank you for the valuable suggestion. We have corrected the sentence: Even in countries with good sanitary conditions, infectious disease threats to kidney health are still essential to be paid attention to. The new sentence could make the paragraph to be more fluent.
Question 2-Line 134: MDC is used, I suggest explaining this acronym when firstly used (multidisciplinary care?)
Answer to question 2: We have offered the full name at the first appearance (Line 132) in the new edition.